# Individualized Approach in the Surgical Management of Hepatocellular Carcinoma: Results from a Greek Multicentre Study

**DOI:** 10.3390/cancers14184387

**Published:** 2022-09-09

**Authors:** Georgios K. Glantzounis, Dimitrios Korkolis, Georgios C. Sotiropoulos, Georgios Tzimas, Anastasia Karampa, Athanasios Paliouras, Alexandros-Georgios Asimakopoulos, Spyridon Davakis, Alexandros Papalampros, Dimitrios Moris, Evangelos Felekouras

**Affiliations:** 1Hepatobiliary and Pancreatic Surgery (HPB) Unit, Department of Surgery, University Hospital of Ioannina, Faculty of Medicine, University of Ioannina, 45500 Ioannina, Greece; 2Department of Surgery, Saint Savvas Hospital, 11522 Athens, Greece; 3Second Propedeutic Department of Surgery, National and Kapodistrian University of Athens, Laikon General Hospital, 11527 Athens, Greece; 4HPB Unit, Department of Surgery, Hygeia Hospital, 15123 Athens, Greece; 5Department of Hygiene and Epidemiology, Faculty of Medicine, University of Ioannina, 45110 Ioannina, Greece; 6First Department of Surgery, National and Kapodistrian University of Athens, Laikon General Hospital, 11527 Athens, Greece

**Keywords:** hepatocellular carcinoma, liver resection, hepatectomy, BCLC criteria, multidisciplinary tumour board meetings

## Abstract

**Simple Summary:**

Hepatocellular carcinoma (HCC) is the most common primary liver cancer with expected increasing frequency in the next few decades. The Barcelona Clinic Liver Cancer (BCLC) Staging System is a widely adopted tool for guiding the therapeutic algorithms of patients with HCC. This classification has been guiding clinical practice for the last two decades. However, emerging data demonstrate that patients beyond the traditional criteria of operability or resectability can benefit from surgical treatment. We present the Greek multicentre experience of treating HCC within and beyond BCLC guidelines.

**Abstract:**

**Background:** Hepatocellular carcinoma (HCC) is the most common primary liver cancer and the third leading cause of death worldwide. The management of HCC is complex, with surgical treatment providing long-term survival in eligible patients. This study aims to present the experience of aggressive surgical management of HCC in Greece. **Methods:** This is a retrospective multicentre clinical study with 242 patients. **Results:** Most patients were male (79%) and had a median age of 71 yrs. According to the most recent BCLC criteria, 172 patients (71.1%) were classified as BCLC 0-A stage, 33 patients (13.6%) were classified as BCLC B, and 37 (15.3%) were classified as BCLC C. A total of 54% of the patients underwent major hepatectomy. Major postoperative morbidity was 15.6%, and the 90-day postoperative mortality rate was 4.5%. The median follow-up was 33.5 months. Three- and five-year overall survival was 65% and 48%, respectively. The median overall survival was 55 months. Significantly, five-year survival was 55% for BCLC A, and 34% and 21% for BCLC B and C, respectively. In univariate analysis, cirrhosis, type of resection (R status), and BCLC stage were associated with overall survival. Multivariate analysis indicated that R1 and R2 resections compared to R0, and BCLC C compared to BCLC 0-A, were independently associated with increased mortality. **Conclusions:** Aggressive surgical treatment of HCC offers satisfactory long-term survival prospects. A significant percentage (29%) of HCCs that underwent liver resection were of the intermediate and advanced BCLC stage. The management of patients with HCC should be discussed in multidisciplinary tumour board meetings on a case-by-case basis to be more effective.

## 1. Introduction

Hepatocellular carcinoma (HCC) is the most common primary liver tumour (90% of tumours). It is also the 5th most common cancer worldwide and the second cause of mortality for men [1,2,3]. HCC usually occurs in the territory of pre-existing liver pathology [3,4]. Risk factors are hepatitis C virus (HCV) and hepatitis B virus (HBV), infection, alcoholic liver disease, aflatoxins, non-alcoholic fatty liver disease (NAFLD), and steatohepatitis (NASH) [5]. Hepatocarcinogenesis is a long process in which multiple molecular mechanisms are implicated [4,6,7,8,9,10]. Although effective antiviral agents for HBV and HCV and effective preventive measures, such as vaccination at birth against HBV infection, are available, the incidence of HCC is increasing [11]. The main reason for this is the increase in obesity and metabolic syndrome, closely related to NAFLD and NASH. NAFLD and NASH are expected to become the first cause of liver cirrhosis and hepatocellular carcinoma in the following decades [12,13].

During the last three decades, numerous staging systems have been proposed for the prognostication and decision-making guidance in HCC. Even though there is no consensus regarding the implementation of one universal staging system, since all of the proposed classifications have deficiencies, the Barcelona Clinic Liver Cancer (BCLC) Staging System remains the most widely used classification system for HCC management guidelines [14]. The main concern around the BCLC Classification is that it groups highly heterogeneous patients under the same stage, recommending treatment modalities that cannot be equally beneficial to all patients [15]. Thus, there is an emerging body of literature identifying patients who are not considered eligible for any surgical intervention with curative intent according to current BCLC recommendations, but can benefit from surgical treatment [16,17,18,19,20,21,22]. 

Furthermore, there have been significant advances over the last five years in the systemic treatment of HCC, as along with the first-line (Sorafenib, Lenvatinib) and second-line therapies (Regorafenib, cabozantinib, or ramucirumab), the use of immunotherapy and anti-angiogenetic therapy have improved overall survival [23,24,25]. Moreover, metronomic therapy has shown promising results [26,27].

This study aims to present the Greek national experience in the surgical management of HCC using a multicentre database of consecutive patients undergoing resection with curative intent over the last 14 years in five hepatobiliary (HPB) centres. This analysis captures the modifications made to BCLC guidelines, how these changes affected the surgical practice, and the current practice and outcomes of surgical management of HCC within and beyond BCLC guidelines. 

## 2. Materials and Methods

This is a multicentre retrospective study on patients with HCC who underwent liver resection from January 2007 to December 2020. The study retrospectively analysed prospectively entered data using the Greek National HPB database. The study included 242 cases from five HPB Centres in Greece, which are, in alphabetical order: Hepatobiliary Centre, Department of Surgery, “Hygeia” Hospital Athens; HPB Unit, Department of Surgery, University Hospital of Ioannina, Ioannina; 1st Surgical Department, Laiko University Hospital, Athens; 2nd Propaedeutic Department of Surgery, Laiko University Hospital, Athens; Department of Surgical Oncology, St. Savvas Oncological Centre, Athens. The primary endpoint was the overall survival (OS) of patients undergoing hepatectomy for HCC in any BCLC stage. Secondary endpoints were postoperative morbidity and mortality. The type of liver resection was classified according to the Brisbane classification [28,29,30]. Postoperative complications were defined using the Dindo–Clavien classification [31]. Tumour burden score (TBS) was defined as the distance from the origin of a Cartesian plane, and calculated by using the maximum tumour diameter and number of tumours from the histopathology report: maximum tumour size (x-axis) and number of tumours (y-axis), so that TBS^2^ = (maximum tumour diameter)^2^ + (number of tumours)^2^ [32,33].

Patients with a radiological diagnosis of HCC were discussed at the multidisciplinary institutional meetings in the presence of a hepatobiliary and transplant-trained surgeon, medical oncologist, radiation oncologist, radiologist, and pathologist. Preoperative staging included computerized tomography (CT) of the chest and abdomen, and liver magnetic resonance imaging (MRI) when indicated. The preoperative screening was performed to determine the pathogenesis of the disease, the preoperative levels of tumour marker (AFP), and the Child-Pugh stage. When a major liver resection (resection of more than three liver segments) [34] was required, liver volumetry was performed, and the risk of postoperative hepatic insufficiency was assessed. In general, less than 40% of residual liver remnant was regarded as high risk for postoperative liver failure, and a right portal vein embolization was performed [35]. In addition, the performance status (PS) score was assessed in all patients.

The BCLC staging system was used to guide the included patients’ decision-making in our analysis. Thus, factors such as the patient’s general condition, the extent, and characteristics of the tumour and liver function were considered. The patients with HCC are classified in five stages: very early-stage disease [stage 0, tumour ≤ 2 cm, preserved liver function, good general condition (PS0)], early-stage disease (stage A, a solitary nodule, or up to 3 each one ≤ 3 cm, preserved liver function, PS 0), intermediate stage patients (stage B, more than two nodules with diameter > 3 cm, preserved liver function, PS 0), advanced stage HCC (C, patients with portal invasion or extrahepatic spread, preserved liver function, PS1–2), and terminal stage (D, end-stage liver function, PS 3–4). The classification was performed according to recent guidelines [36,37,38] and with each period’s existing guidelines [14,39]. Subsequently, early, intermediate, and selected advanced stage patients (mainly with portal vein invasion) underwent hepatectomy with an individualized decision. Intraoperative ultrasound was performed on all patients. The duration of hospitalization, immediate postoperative complications, and 90 days mortality was recorded. Upon histological examination, patients were reassessed on a regular external basis, based on a follow-up protocol by the HPB surgical team of each unit. The follow-up included history, clinical examination, liver function tests, AFP levels, chest, abdomen, and pelvis CT every four months for the first two years, and then every six months. Recurrence of the disease was diagnosed and treated accordingly (hepatic resection, liver transplantation, embolization, ablation, systemic therapy) considering the extent of the disease and the patient’s performance status. 

### Statistical Analysis

Patients’ demographic, clinical, and pathological characteristics are described overall and stratified according to the BCLC Classification as frequency rates and percentages. Overall survival rates were evaluated by the Kaplan–Meier method and were compared using the log-rank test. Univariate analysis was performed to estimate the association of age, sex, cirrhosis, hepatitis B virus, hepatitis C virus, alcohol, non-alcoholic fatty liver disease, R0 resection vs. R1-R2 resection, tumour burden score, and BCLC Classification with mortality. All the significant variables were introduced into a multivariate Cox proportional hazards model; *p*-value < 0.05 was considered statistically significant. All statistical analyses were performed using the STATA 17 (StataCorp LP, College Station, TX, USA) software.

## 3. Results

### 3.1. Patients’ Demographic, Clinical, and Pathological Characteristics

Two hundred and forty-two patients were included in the study, with 79% of them being male. The median age was 71 years (range 21–89). In 52% of the patients, HCC was developed in the setting of a cirrhotic liver. Regarding the aetiology of HCC, 34.7% of the patients had HBV infection, 15.3% had HCV infection, while alcoholic liver disease (ALD) history was present in 46% of the study’s patients. Finally, NAFLD was found in 33% of cases. In 4% of the patients with HCC, the etiopathogenesis was due to other factors. The majority of the patients were Child–Pugh A (73%), while the remaining 27% were Child B. Table 1 presents the demographic, clinical, and pathological characteristics of patients stratified according to the BCLC Classification. 

Regarding the BCLC Classification, the most recent criteria were used. Of the patients, 172 (71.1%) were BCLC A, 33 patients (13.6%) were BCLC B, and 37 patients (15.3%) were BCLC C (Table 1). If we used the BCLC criteria according to the period they were implemented, then the classification is different: 106 patients would be classified as BCLC A (43.8%), 99 patients as BCLC B (41%), and 37 as BCLC C (15.2%).

In summary, the majority of the patients were male (79%), HBV infection and NAFLD were the main causes of chronic liver disease, and 70% of the patients were BCLC A according to the most recent criteria. 

### 3.2. Type of Liver Resections and Histopathology Results

A total of 131 patients (54%) underwent major hepatectomy while 111 patients (46%) underwent minor hepatectomy. Laparoscopic liver resection was performed in 13 patients (5.4% of liver resections) (Table 2). 

Based on the pathological examination of the liver tissue, 126 patients (52%) had pre-existing cirrhosis, and 73% were R0 resections. The median tumour diameter was 7.37 cm (range 1–25 cm) (Table 3).

In summary, the majority of patients (54%) underwent major liver resection, while liver cirrhosis was found in 52% of them.

### 3.3. Postoperative Outcomes and Survival

The median length of hospitalization was 11 days (range 3–100 days). Forty per cent of the patients developed postoperative complications, whereas 24.4% had a minor complication according to the Clavien–Dindo score, while 15.6% had a major complication (Dindo–Clavien class III. IV). The 90-day mortality was 4.5%. The remaining patients underwent regular follow-ups. The median follow-up was 33.5 months (range 1–146 months). The median overall survival time was 55 months (95% confidence interval: 46–70 months). The 1-, 3-, and 5-year overall survival rates were 89%, 65%, and 48%, respectively (Figure 1). Concerning BCLC Classification, there was a significant survival difference between the three categories (Figure 2; *p* < 0.001). The overall survival at 1, 3, and 5 years was 95%, 77%, and 55%, respectively for BCLC 0-A; 74%, 42%, and 34%, respectively for BCLC B; 70%, 25%, and 21%, respectively for BCLC C (Table 2). There was also a significant survival difference based on cirrhosis status (Figure 3; *p* = 0.034). Specifically, for cirrhotic patients, the 1-, 3-, and 5-year overall survival were 86%, 61%, and 43%, respectively and for non-cirrhotic were 92%, 70%, and 53%, respectively (Table 4). The recurrence rate was 42% in 3 years and 56% in 5 years. The recurrence rate in 5 years was 47% for BCLC A, 59% for BCLC B, and 89% for BCLC C.

In summary, major complications were noted in 16% of patients and the 90-day mortality was 4.5%. The median survival was 55 months, and the 5-year overall survival was 48%. BCLC B and C stage and the presence of cirrhosis had a negative effect on overall survival.

### 3.4. Univariate and Multivariate Analysis

In univariate analysis, cirrhosis, R type of resection, and BCLC stage were associated with overall survival. Multivariate Cox model indicated that R1-R2 resection compared to R0 (HR: 2.43, 95% CI: 1.63 3.64, *p* = 1.5 × 10^−5^) and BCLC C compared to BCLC 0-A (HR: 2.54, 95%CI: 1.53–4.21, *p* = 3.3 × 10^−4^) were independently associated with increased mortality (Table 4).

## 4. Discussion

This study presents long-term outcomes of patients with HCC undergoing liver resection within and beyond BCLC guidelines. Our data support the role of surgical resection with curative intent in selected patients with BCLC B achieving a 5-year survival rate of 34% and a median survival of 32 months. The current BCLC recommendations suggest mainly embolization (TACE) for these patients, achieving a median survival of 18–27 months [36,40]. A recent phase II clinical trial showed that TACE in patients with intermediate-stage HCC has an OS of 26 months and 3- and 5-year survival of 36% and 2.7%, respectively [41]. 

The BCLC staging system is the most popular for prognosis and therapeutic guidance in patients with HCC [42]. However, it is considered to be very restrictive regarding indications for surgical management, including resection and transplantation, and for this reason, it has been heavily questioned by the HPB community [20,43,44,45,46,47,48].

The current study has used more liberal selection criteria in the surgical management of patients with HCC. Liver resection has been applied at the early stage as Greece is a country where organ availability for liver transplantation is limited [49,50]. It has also been applied for HCCs in intermediate and advanced stages, in 29% of the cohort of patients, providing satisfactory long-term survival and good quality of life. Significantly, when we adjusted our analysis by applying the contemporary BCLC criteria (BCLC criteria used at the time of liver resection), we found that the majority of patients (56%) were in the intermediate (41%) and advanced stage (15%). Advanced stage (BCLC-C) includes a very heterogeneous group of patients: portal invasion, lymph node or distal metastases, and decreased performance status (PS 1–2). According to BCLC criteria, systemic targeted therapy is the standard of care with expected progression-free survival of 6.8 months, and overall survival of <20 months [23,51]. However, patients with limited portal vein invasion (pV1 or pV2) or responding well to neo-adjuvant treatment could benefit from surgical resection [22]. 

More importantly, the present study shows that the Greek HPB centres have an aggressive approach in patients with HCC, as it is reflected by the median diameter of the tumours resected (7.37 cm), the high percentage of major hepatectomies (54%), and the fact that 27% of these patients were Child B. Furthermore, the indications were outside the BCLC criteria in more than 50% of the liver resections for HCC, if the existing criteria were applied at the time of operation. Our R0 resection rate is relatively low compared to other studies [20,22], and the high percentage of patients with advanced HCCs (tumours > 10 cm, BCLC B and C) could partially explain this finding. Despite these unfavourable parameters, a good 5-year OS of 48% was achieved. The selection criteria used and the long-term results achieved are in accordance with the results of large multicentre studies [20,52,53,54], and comparable with the results of recent systematic reviews [21,22]. A recent systematic review and meta-analysis showed that anatomic liver resections offer better overall and disease-free survival [55]. This finding could partially explain our long-term results, as most patients underwent major anatomic resections.

Surrogates of HCC behaviour, such as tumour burden score (TBS), and patient clinical performance, such as Model for End-Stage Liver Disease (MELD) score, can identify patients who benefit from surgical management beyond traditional criteria. These nuances have not been captured in the updated BCLC guidelines published in 2022 [38]. More specifically, recent multicentre data support a beneficial role of surgery in patients with BCLC B/C; despite the fact that they have a higher risk for early (<2 years) or multiple intrahepatic recurrences compared to BCLC 0/A (*p* = 0.011), and shorter 5-year OS (51.6% for BCLC B/C versus 76.9% for BCLC 0/A, *p* = 0.003), half of these patients can survive for five years after resection, a finding that is above any expectation from other recommended treatment (TACE and sorafenib) [56]. Similarly, in patients with multinodular HCC undergoing resection, those with low TBS achieved a 73.7% 5-year OS survival, whereas patients with high TBS had only 13.1% (*p* < 0.001). This highlights how tumour burden can refine the management of patients with presumably unresectable disease [57]. In our study, low TBS was associated with better survival. However, in the univariate analysis, there were no statistically significant differences between the three groups. This finding could be due to the relatively small number of patients in the present study. 

Finally, a machine-learning analysis showed that factors such as comorbidities and high pre-resection AFP, as well as post-resection factors such as TBS and lymphovascular invasion, could be the best predictors of OS in patients with BCLC-A, and TBS was the single best predictor of outcomes in patients with BCLC-B undergoing resection for HCC [58]. 

A recent randomized phase II trial showed that the perioperative use of immunotherapy in resectable HCCs is safe and feasible. Immunotherapy as neo-adjuvant therapy in advanced HCC may contribute, in combination with surgery, to better long-term survival [59].

The present study, along with emerging literature, supports an individualized approach for the surgical management of patients with HCC based on clinical performance status, satisfactory baseline liver function (Child A or B, total bilirubin < 2 mg /dL, INR < 2, platelets > 80,000), adequate future live remnant (>40% in the presence of cirrhosis), in BCLC 0-A patients, as well as selected cases of BCLC B and C. The decision should be taken at the multidisciplinary tumour board level. We are happy to see that the recent update to BCLC guidelines proposes a personalized HCC treatment approach where the tumour board should choose the option which provides the best survival [38].

Another interesting finding of our study is that liver resection is safe for elderly patients (>75 yrs) and leads to long-term survival rates similar to those of younger patients [60,61].

In the near future, there is a need to identify reliable prognostic markers to help us choose the best treatment option for each patient [62]. Furthermore, novel treatments such as immunotherapy and anti-angiogenetic therapy should have a role in the neo-adjuvant setting in choosing a radical surgical treatment (resection or transplantation) for the patients who respond well to the neo-adjuvant therapies. 

The present study has several strengths and weaknesses that should be mentioned. First, the study captures more than ten years of HPB practice in both academic and private practice settings. Moreover, the patient population was homogeneous, thus limiting the potential bias related to different racial backgrounds. Moreover, Greek HPB surgeons have demonstrated adaptability to novel technologies that facilitate the safe, bloodless, and efficient performance of major hepatectomies, including ablation devices [63,64,65] and novel techniques [66,67] despite the limited resources, such as ICU availability [68,69]. As far as weaknesses are concerned, this is a retrospective study, and as such it is subject to bias. The number of patients included is relatively low, in order to have a clear conclusion regarding predictive markers such as TBS or selection criteria for liver resection in BCLC B and C patients. Moreover, the percentage of laparoscopic liver resections in this study was low, and this is an area which requires improvement in the near future, as significant advances have taken place in the era of laparoscopic and robotic surgery in recent years and patients with HCC have a significant benefit in postoperative morbidity with these approaches [62,70,71,72,73,74]. 

## 5. Conclusions

In conclusion, this study shows that a significant percentage of patients with HCC managed surgically in Greece are of intermediate and advanced stage, or Child B, and require major liver resections. Through an individualized approach, good long-term results have been achieved. Further prospective studies are required to clarify the subgroups of patients with intermediate or advanced stage HCC who will benefit from liver resection.

## Figures and Tables

**Figure 1 cancers-14-04387-f001:**
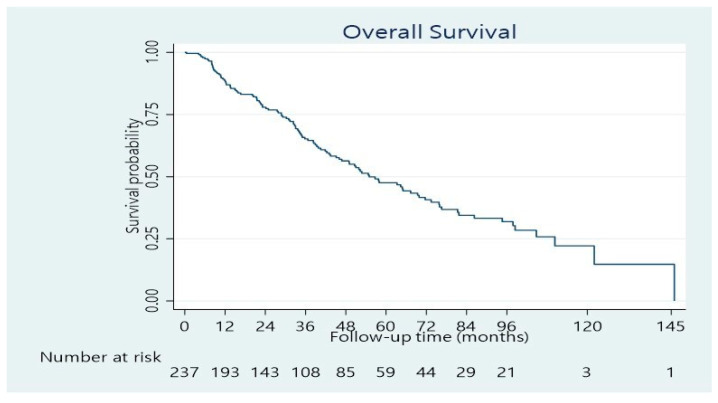
Overall survival distribution of patients resected for HCC.

**Figure 2 cancers-14-04387-f002:**
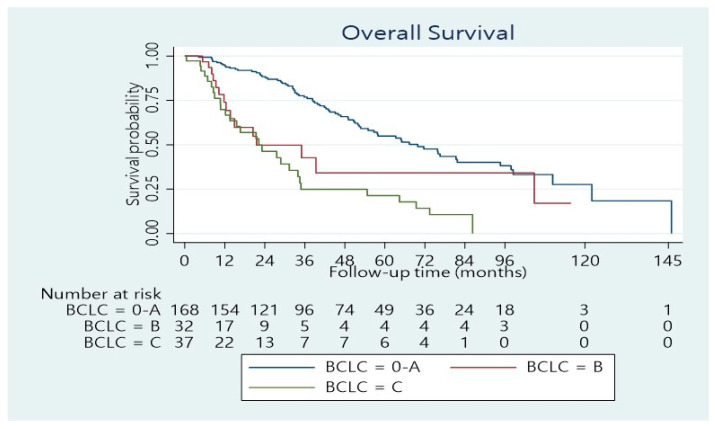
Overall survival distribution stratified according to the BCLC stages (*p* < 0.001).

**Figure 3 cancers-14-04387-f003:**
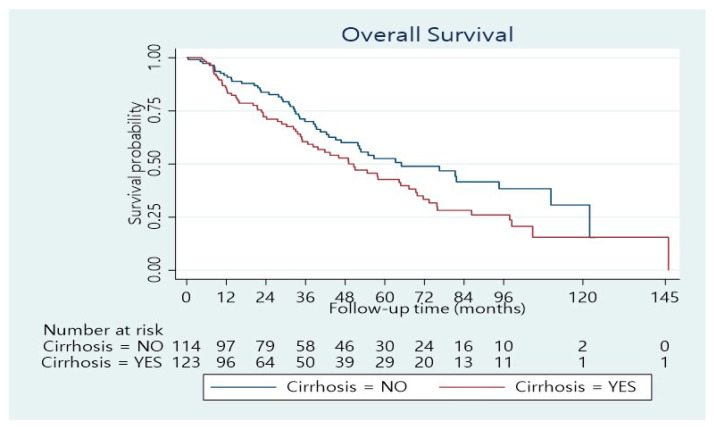
Overall survival distribution stratified according to Cirrhosis status (*p* = 0.034).

**Table 1 cancers-14-04387-t001:** Demographic, clinical, and pathological characteristics of patients stratified according to the BCLC classification.

Characteristics	Overall (*n* = 242)	BCLC 0-A (*n* = 172)	BCLC B (*n* = 33)	BCLC C (*n* = 37)
Age, *n* (%)				
<75 years	165 (68.18)	110 (63.95)	27 (81.82)	28 (75.68)
≥75 years	77 (31.82)	62 (36.05)	6 (18.18)	9 (24.32)
Sex, *n* (%)				
Male	191 (78.93)	133 (77.33)	25 (75.76)	33 (89.19)
Female	51 (21.07)	39 (22.67)	8 (24.24)	4 (10.81)
HBV, *n* (%)				
No	158 (65.29)	113 (65.70)	20 (60.61)	25 (67.57)
Yes	84 (34.71)	59 (34.30)	13 (39.39)	12 (32.43)
HCV, *n* (%)				
No	205 (84.71)	146 (84.88)	26 (78.79)	33 (89.19)
Yes	37 (15.29)	26 (15.12)	7 (21.21)	4 (10.81)
Alcohol, *n* (%)				
No	130 (53.72)	99 (57.56)	15 (45.45)	16 (43.24)
Yes	112 (46.28)	73 (42.44)	18 (54.55)	21 (56.76)
NAFLD, *n* (%)				
No	162 (66.94)	111 (64.53)	25 (75.76)	26 (70.27)
Yes	80 (33.06)	61 (35.47)	8 (24.24)	11 (29.73)
Cirrhosis, *n* (%)				
No	116 (47.93)	92 (53.49)	12 (36.36)	12 (32.43)
Yes	126 (52.07)	80 (46.51)	21 (63.64)	25 (67.57)
TBS, *n* (%)				
Low	27 (11.16)	25 (14.53)	1 (3.03)	1 (2.70)
Medium	196 (80.99)	137 (79.65)	29 (87.88)	30 (81.08)
High	19 (7.85)	10 (5.81)	3 (9.09)	6 (16.22)
R, *n* (%)				
R0	159 (73)	129 (81.2)	19 (70.4)	11 (34.4)
R1-R2	59 (27)	30 (18.8)	8 (29.6)	21 (65.6)
Death, *n* (%)				
No	120 (50.63)	94 (55.95)	17 (53.13)	9 (24.32)
Yes	117 (49.37)	74 (44.05)	15 (46.88)	28 (75.68)

**Table 2 cancers-14-04387-t002:** Type of resection.

Liver Resection	Total	BCLCA	BCLCB	BCLCC
Major	131 (54.1%)	95	17	19
Right hepatectomy	75 (57.2%)	54	10	11
Left hepatectomy	49 (37.4%)	36	5	8
Central hepatectomy	7 (5.4%)	5	2	0
Minor	111 (45.9%)	77	16	18
1 segment	57 (51.3%)	44	6	7
2 segments	54 (48.4%)	33	10	11
Laparoscopic hepatectomy	13 (5.4%)	10	0	3
Major	2	1	0	1
Minor	11	9	0	2

**Table 3 cancers-14-04387-t003:** Histopathology results.

Liver Background	Total	BCLC 0-A	BCLC B	BCLC C
**Total**	242	172	33	37
**Cirrhosis**	126 (52%)	80 (46.5%)	21 (63.6%)	25 (67.5%)
**Normal liver /Fibrosis**	116 (48%)	92(53.5%)	12 (36.4%)	12 (32.5%)
**Single**	204	172	0	32
<2 cm	11	11	0	0
2–5 cm	60	58	0	2
5–10 cm	90	72	0	18
>10 cm	43	31	0	12
**Multinodular**	38	0	33	5
*n* = 2	24	0	21	3
*n* = 3	9	0	8	1
*n* > 3	5	0	4	1
**R status, *n* (%)**				
R0	159 (73%)	129 (81.2%)	19 (70.4%)	11 (34.4%)
R1–R2	59 (27%)	30 (18.8%)	8 (29.6%)	21 (65.6%)

**Table 4 cancers-14-04387-t004:** Univariate and multivariate analysis of overall survival.

Variable		1-Yr Survival (%)	3-Yr Survival (%)	5-Yr Survival (%)	Overall Median Survival (Months)	Univariate Analysis			Multivariate Analysis		
						HR	95 CI	*p*-Value	HR	95 CI	*p*-Value
Age < 75		88	63	47	53						
≥75		91	70	48	58	0.97	0.66–1.43	0.886			
Sex	MALE	88	65	48	57						
	FEMALE	93	65	48	55	0.95	0.60–1.50	0.830			
Cirrhosis	NO	92	70	53	65						
	YES	86	61	43	51	1.48	1.02–2.15	**0.036**	1.19	0.78–1.82	0.416
HBV	NO	90	64	49	55						
	YES	87	67	46	52	1.04	0.71–1.52	0.836			
HCV	NO	89	66	50	58						
	YES	88	61	38	42	1.40	0.88–2.24	0.153			
ALCOHOL	NO	88	68	46	55						
	YES	90	61	51	63	1.13	0.78–1.64	0.521			
NAFLD	NO	86	63	47	52						
	YES	94	70	49	57	0.87	0.59–1.30	0.499			
R	R0	94	79	57	76						
	R1–R2	74	36	27	29	2.99	2.04–4.38	**1.77 × 10^−8^**	2.43	1.63–3.64	**1.5 × 10^−5^**
TBS	Low	93	65	58	82						
	Medium	88	65	46	55	1.57	0.79–3.11	0.194			
	High	88	62	49	52	1.75	0.67–4.55	0.252			
BCLC	0–A	95	77	55	70						
	B	74	42	34	32	2.14	1.22–3.76	**0.008**	1.79	0.95–3.35	0.071
	C	70	25	21	22	3.55	2.28–5.52	**1.96 × 10^−8^**	2.54	1.53–4.21	**3.3 × 10^−4^**

## Data Availability

The data can be shared up on request.

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
