# Peer review of "Individualized Approach in the Surgical Management of Hepatocellular Carcinoma: Results from a Greek Multicentre Study"

_cancers, 2022, doi:10.3390/cancers14184387_

Round 1

Reviewer 1 Report

The article entitled "Individualized Approach in The Surgical Management of Hepatocellular Carcinoma: Results from A Greek Multicenter Study" provides a brief Introduction and addresses the interesting and hot topic of personalized medicine but then the Results section is a mere compilation of results without any conclusion, link or mechanistical insight. 

Furthermore, MELD and ABIC scores are not mentioned at all and other interesting analysis, such as those related to pharmacological treatment, age, sex and other factors, are not performed.

I think this article could be published in a pure clinical journal.

Author Response

Reviewer 1:
Comment 1: The article entitled "Individualized Approach in The Surgical Management of Hepatocellular Carcinoma: Results from A Greek Multicenter Study" provides a brief Introduction and addresses the interesting and hot topic of personalized medicine but then the Results section is a mere compilation of results without any conclusion, link or mechanistical insight.

Response: - This multi-centre trial on the surgical management of Hepatocellular carcinoma (HCC) aims to point out that the surgical management of hepatocellular carcinoma should be discussed case by case in the multi-disciplinary tumour boards in order for each patient to get the maximum benefit. This approach is in line with the updated guidelines of BCLC group (J Hepatology, 2022;76:681-693), which conclude: <<critical insight and expert knowledge are required to make clinical decisions for individual patients, considering all of the parameters that must be considered to deliver personalized clinical management>>.

Relevant conclusions are included in the discussion section.

Comment 2: Furthermore, MELD and ABIC scores are not mentioned at all and other interesting analysis, such as those related to pharmacological treatment, age, sex and other factors, are not performed.

Response: MELD score and ABIC score are not mentioned in the paper as they were not used in selecting patients with HCC for liver resection. However, tumour burden score (TBS) was used, and univariate and multivariate analysis of patient’s demographic, clinical and pathologic characteristics was performed in the revised manuscript.

Comment 3: I think this article could be published in a pure clinical journal.

Response: We believe that the doctors of different medical specialities, dealing with the management of hepatocellular carcinoma, should speak a <<common>> language as the management of HCC is complex and should be done, on a personalized basis, after discussion in the multi-disciplinary tumour boards. For this reason, it is important the basic principles of the surgical management of HCC be published in a non-surgical Journal.

Reviewer 2 Report

Dear Editor, thank you so much for inviting me to revise this manuscript about HCC.

This study addresses a current topic.

The manuscript is quite well written and organized. English could be improved.

Figures and tables are comprehensive and clear.

The introduction explains in a clear and coherent manner the background of this study.

We suggest the following modifications:

  • Introduction section: although the authors correctly included important papers in this setting, we believe some recent studies regarding novel treatments and topics in HCC management should be cited within the introduction ( PMID: 34429006; PMID: 34764464; PMID: 29968763 ), only for a matter of consistency. We think it might be useful to introduce the topic of this interesting study.
  • Methods and Statistical Analysis: nothing to add.
  • Discussion section: Very interesting and timely discussion. Of note, the authors should expand the Discussion section, including a more personal perspective to reflect on. For example, they could answer the following questions – in order to facilitate the understanding of this complex topic for readers: what potential does this study hold? What are the knowledge gaps and how do researchers tackle them? How do you see this area unfolding in the next 5 years? We think it would be extremely interesting for the readers.

However, we think the authors should be acknowledged for their work. In fact, they correctly addressed an important topic, the methods sound good and their discussion is well balanced.

One additional little flaw: the authors could better explain the limitations of their work, in the last part of the Discussion.

We believe this article is suitable for publication in the journal although major revisions are needed. The main strengths of this paper are that it addresses an interesting and very timely question and provides a clear answer, with some limitations.

We suggest a linguistic revision and the addition of some references for a matter of consistency. Moreover, the authors should better clarify some points.

Author Response

Comment 1: This study addresses a current topic. The manuscript is quite well written and organized. English could be improved.

Response: The English language has been improved.

Comment 2: Figures and tables are comprehensive and clear.
The introduction explains in a clear and coherent manner the background of this study.

We suggest the following modifications:

• Introduction section: although the authors correctly included important papers in this setting, we believe some recent studies regarding novel treatments and topics in HCC management should be cited within the introduction (PMID: 34429006; PMID: 34764464; PMID: 29968763 ), only for a matter of consistency. We think it might be useful to introduce the topic of this interesting study.

Response: The proposed papers have been added, and the novel treatments are included in the introduction now.

Comment 2:
Methods and Statistical Analysis: nothing to add.

Discussion section: Very interesting and timely discussion. Of note, the authors should expand the Discussion section, including a more personal perspective to reflect on. For example, they could answer the following questions – in order to facilitate the understanding of this complex topic for readers: what potential does this study hold? What are the knowledge gaps and how do researchers tackle them? How do you see this area unfolding in the next 5 years? We think it would be extremely interesting for the readers.

Response: The discussion has been expanded, and our perspective on the future management of HCC is included.

Comment 3: However, we think the authors should be acknowledged for their work. In fact, they correctly addressed an important topic, the methods sound good and their discussion is well balanced.
One additional little flaw: the authors could better explain the limitations of their work, in the last part of the discussion.
We believe this article is suitable for publication in the journal although major revisions are needed. The main strengths of this paper are that it addresses an interesting and very timely question and provides a clear answer, with some limitations the manuscript.

Response: Limitations of our work have been better explained in the discussion section.

Reviewer 3 Report

In this manuscript, the Authors performed a retrospective analysis of demographic-clinical-pathological characteristics, perioperative and oncologic outcomes of patients who underwent liver resection for HCC at four Greek HPB centers, during a 13 years period. 

Patients included in the current study were classified, according to the most recent BCLC criteria, in three groups of patients, who showed different characteristics and outcomes (short- and long-term). Patients in BCLC C group were more often affected by more advanced disease and had shorter survivals. 

The authors state the the results of this manuscript support the results of previous studies highlighting the inability of current BCLC criteria in adequately selecting the optimal management modality for patients in stage B or C: while the BCLC criteria suggests a non surgical management, an increasing clinical evidence suggests that some of them may benefit of a liver resection. 

some comments are due: 

  •   The manuscript needs to be reviewed by a mother tongue English scientific reviewer, because the English stuyle is very poor, the text contains many grammatical or ortographic  mistakes (mainly missing verbs) and many sintax errors.

concerning the study results, i think that the author should work on the tables: 

  • the tables contain some numeric inaccuracies, which should be corrected: for example in table 2 the sum of patients undergoing different kind of liver resection in the groups BCLC-A and BCLC -C do not correspond to the total number of patients for each group of patients shown in table 1 (line: number of patients): please correct. 
  • again in the table 2 the percentages into parentesis for major liver resection and minor liver resection according to BCLC stages should be calculated using as a denominator the number of patients in the respective BCLC group (namely 151, 34, and 25). Please correct. 
  • I suggest the authors to compare characteristics and outcomes of the three groups of patients (BCLC-A vs BCLC-B vs BCLC-C): such comparison may add to the manuscript significance. 
  • I suggest to add, among data, some information usually reported in order to surrogate tumor burden, like the need for preoperative PVO or TACE, or surgical difficulty, like operation duration, intraoperative blood loss, use and duration  of pedicle clamping, intraoperative need for transfusion.  
  • I suggest the authors to show in a table the perioperative outocmes (including morbidity and mortality) according to BCLC. 
  • I suggest to merge data (reported in multiple tables) in not more than two tables (for example: one showing clinical-demographic and surgical intraoperative data, the other showing pathological data and postoperative outcomes). 

I suggest the authors to report long term survival in an additional table, in order to allow to the readers a rapid overview of survivals in a numerical representation, and to merge all the kaplan-maeyer curves in a single figure. 

again, concerning long-term survivals a comparison of survivals according to BCLS stage and to the presence of cirrohosi may add to the manuscript. 

I also suggest the authors to investigate factors associated with long term survivals: this could add points for comparisons with available data and for a more interesting discussion.

The discussion enlists many data coming from previously published manuscripts, however the absence of many of such data in the current manuscript make eventual comparisons really difficult or impossible. The  author should try to interpret their results and compare them with existing data, commenting them. 

Author Response

Reviewer 3:

Comment 1: The manuscript needs to be reviewed by a mother tongue English scientific reviewer because the English style is very poor, the text contains many grammatical or orthographic mistakes (mainly missing verbs) and many syntax errors.

Response: The English language has been checked, and grammatical, and syntax errors have been corrected.

Comment 2:
concerning the study results, I think that the author should work on the tables:
• the tables contain some numeric inaccuracies, which should be corrected: for example in table 2 the sum of patients undergoing different kind of liver resection in the groups BCLC-A and BCLC -C do not correspond to the total number of patients for each group of patients shown in table 1 (line: number of patients): please correct.
• again in the table 2 the percentages into parenthesis for major liver resection and minor liver resection according to BCLC stages should be calculated using as a denominator the number of patients in the respective BCLC group (namely 151, 34, and 25). Please correct.

Response: Tables have been corrected.

Comment 3:
• I suggest the authors to compare characteristics and outcomes of the three groups of patients (BCLC-A vs BCLC-B vs BCLC-C): such comparison may add to the manuscript significance.

Response: A comparison of the three groups has been done (table 1, table 4, fig. 2)

Comment 4:
• I suggest to add, among data, some information usually reported in order to surrogate tumour burden, like the need for preoperative PVO or TACE, or surgical difficulty, like operation duration, intraoperative blood loss, use and duration of pedicle clamping, intraoperative need for transfusion.

Response: The tumour burden score has been calculated and is included in the analysis. Regarding preoperative management and operative data have not been included as there were several missing values, and we felt that the presentation of these data would not offer in the conclusions of the paper.

Comment 5:
• I suggest the authors to show in a table the perioperative outcomes (including morbidity and mortality) according to BCLC.

Response: Perioperative outcomes are included in the text for technical reasons.

Comment 6:
• I suggest to merge data (reported in multiple tables) in not more than two tables (for example: one showing clinical-demographic and surgical intraoperative data, the other showing pathological data and postoperative outcomes).

Response: tables have been merged. However, due to differences in data analysis four tables have been included.

Comment 7:
I suggest the authors to report long term survival in an additional table, in order to allow to the readers a rapid overview of survivals in a numerical representation, and to merge all the Kaplan-Meyer curves in a single figure. Αgain, concerning long-term survivals a comparison of survivals according to BCLS stage and to the presence of cirrhosis may add to the manuscript.

Response: Survival data between groups is presented in table 4. Kaplan Meyer curves analysis has been done, as long as a comparison between groups. Analysis of survival in the presence of cirrhosis has also been performed (fig. 3). However, we felt that the use of three figures represents better the results of our study.

Comment 8:
I also suggest the authors to investigate factors associated with long term survivals: this could add points for comparisons with available data and for a more interesting discussion.

Response: Univariate and multivariate analysis of study data has been performed, and the results are discussed in the discussion section.

Comment 9:
The discussion enlists many data coming from previously published manuscripts, however, the absence of many such data in the current manuscript make eventual comparisons really difficult or impossible. The author should try to interpret their results and compare them with existing data, commenting them.

Response: Further interpretation and comparison of the study data with other studies has been made in the discussion section.

Reviewer 4 Report

Interesting paper, contains important information about the possibility of exceeding the limits of BCLC scale when qialifying patients with HCC for liver resection, especialy when the cancer is caused not by cirrhosis but by liver fibrosis

Author Response

Reviewer 4:

Comments: Interesting paper, contains important information about the possibility of exceeding the limits of BCLC scale when qualifying patients with HCC for liver resection, especially when the cancer is caused not by cirrhosis but by liver fibrosis.

Response: We thank the reviewer for his comments
Furthermore, the material of our work has not been previously published or submitted elsewhere for publication and will not be sent to another journal until a decision is made concerning publication by your journal. All authors have read and approved the revised manuscript and take public responsibility for it. Finally, all authors wish to state that they have no conflict of interest of any kind and that they received no outside support.
I look forward to your response at your earliest convenience.

Yours sincerely,

Georgios Glantzounis, MD, PhD, FEBS
Professor of Surgery and Transplantation
Medical school, University of Ioannina
Corresponding author